# The Role of Combined Muscle Ultrasound and Bioimpedentiometry Parameters for Sarcopenia Diagnosis in a Population of Hospitalized Older Adults

**DOI:** 10.3390/nu16152429

**Published:** 2024-07-26

**Authors:** Alfredo Zanotelli, Andrea P. Rossi, Letizia Del Monte, Gianluca Vantini, Giovanni Stabile, Silvia Urbani, Anna Giani, Elena Zoico, Alessio Babbanini, Francesco Fantin, Mauro Zamboni, Gloria Mazzali

**Affiliations:** 1Section of Geriatric Medicine, Ospedale Santa Maria del Carmine, 38068 Rovereto, Italy; alfredo.zanotelli@outlook.com; 2Division of Geriatrics, Department of Medicine, Ospedale Cà Foncello, 31100 Treviso, Italy; 3Department of Medicine, Geriatrics Division, University of Verona, 37100 Verona, Italy; letizia.delmonte@aovr.veneto.it (L.D.M.); gianluca.vantini@univr.it (G.V.); stabile.giovanni.1993@gmail.com (G.S.); silviaurbani93@gmail.com (S.U.); annagiani92@gmail.com (A.G.); elena.zoico@univr.it (E.Z.); mauro.zamboni@univr.it (M.Z.); gloria.mazzali@uniivr.it (G.M.); 4Division of Geriatrics, Department of Medicine, Dentistry, Pediatric and Gynecology, Healthy Aging Center, University of Verona, 37126 Verona, Italy; alessiobabba@gmail.com; 5Division of Geriatrics, University of Trento, 38100 Trento, Italy; francesco.fantin@univr.it

**Keywords:** muscle, muscle ultrasound, bioimpedentiometry, muscle quality, phase angle, myosteatosis, sarcopenia

## Abstract

Background: For the study of quantitative and qualitative muscle parameters, ultrasound and bioelectric impedance analysis are reliable, non-invasive, and reproducible. The aim of this study was to test the combined role of those techniques for the diagnosis of sarcopenia in a population of hospitalized older males and females. Methods: A total of 70 subjects were recruited, including 10 healthy adults and 60 hospitalized elderly patients with a good level of independence and cooperation, with and without sarcopenia. The rectus femoris cross-sectional area (CSA), thickness, echogenicity, and compressibility were measured with ultrasound echography. The phase angles (PhAs) and skeletal muscle mass were calculated by bioimpedence analysis. The muscle quality index (MQI) was calculated as the product of CSA and PhA. Results: Muscle compressibility was greater and PhA was lower in sarcopenic when compared with non-sarcopenic subjects. The threshold values for sarcopenia diagnosis in both sexes of CSA, of PhA, and of the MQI were identified. The obtained CSA values showed an AUC of 0.852 for women and 0.867 for men, PhA of 0.792 in women and 0.898 in men, while MQI was 0.900 for women and 0.969 for men. Conclusions: The newly calculated cut-off values of CSA, PhA, and MQI predicted the presence of sarcopenia with good sensitivity and specificity values. The use of the MQI proved to be more promising than the separate use of CSA and PhA in both male and female subjects.

## 1. Introduction

Sarcopenia is a progressive and generalized musculoskeletal disorder that is associated with an increased likelihood of adverse outcomes, including falls, fractures, physical disability, and mortality [1].

In the latest guidelines proposed by the European Working Group on Sarcopenia (EWGSOP), muscle strength is considered the best predictor of adverse outcomes. Furthermore, both muscle mass and muscle quality have been shown to be compromised [2,3,4,5], ultimately confirming the diagnosis of sarcopenia. The EWGSOP guidelines recognize bioelectric impedance analysis (BIA), along with dual-energy X-ray absorptiometry, computer tomography, and magnetic resonance, as a reference method for sarcopenia diagnosis [2].

BIA is a reliable, non-invasive, and reproducible method for muscle mass quantification; conversely, ultrasound has recently attracted interest as a method for quantitative and qualitative muscle parameter measurements [6,7,8]. Many studies demonstrate high concordance with gold standard radiology methods together with recent biopsy studies, and it has the advantage of being time-saving and low-cost, while having no contraindications [6,7,8].

Ultrasound is a widely used technique to measure both muscle mass by measuring the cross-sectional area (CSA) and muscle quality, but it is still not recognized as a criterion method for sarcopenia diagnosis by the EWGSOP [2,9]. This method has good intra- and interobserver reliability, even when used on older subjects [9]. Among the ultrasound methods used, the CSA measurement of the quadriceps muscles is the most frequent and reliable. Studies comparing ultrasound with computed tomography (CT) methods have shown a decline in CSA of the lower limb muscles in adulthood with good concordance between the two methods [10,11]. Furthermore, among the thigh muscles, there is a temporal discrepancy between the anterior (mainly rectus femoris) and posterior (biceps femoris) muscles, seeing as the former is primarily involved in age-related muscle decline [12].

The term “muscle quality” refers to the biochemical and cellular composition that is capable of influencing, together with mass, contractile efficiency, resistance to effort, and anabolic response. However, there is currently no universally accepted definition for muscle quality. Changes in muscle quality can contribute to loss of muscle function regardless of mass and lead to the onset of sarcopenia [13].

Recent studies have shown that muscle ultrasound can be used to measure muscle quality parameters [14,15]. For instance, muscle echogenicity is an indirect qualitative parameter and is linked to biological modifications of the muscle that can influence contractile efficiency and physical performance. Various structural changes in the muscle such as fatty infiltration, fibrosis, and edema produce increased echogenicity. However, muscle ultrasound is not able to distinguish these different types of components. Studies on muscle biopsies from dogs have shown a correlation between hyperechogenicity and the degree of interstitial fibrosis [14], as well as the degree of intra- and intermuscular fat [15].

Another qualitative ultrasound parameter is stiffness, a mechanical property of muscle linked to the density of molecules of which muscle is composed, and it is capable of expressing itself in different phenotypes: from trigger points to density and fibrosis.

BIA is the measurement of bioelectrical impedance in biological tissues. BIA is based on the principle that biological tissues behave as conductors, semiconductors, dielectrics, or insulators. The intra- and extracellular electrolyte solutions of soft tissues, in particular lean tissues, are excellent conductors, while bone and lipid droplets are not influenced by the currents used in clinical plethysmographs. Therefore, BIA can only analyze soft tissue parts. Fat mass is then estimated from the difference in lean mass and body weight [12,16,17,18]. Multiple studies have been carried out to determine equations that calculate appendiceal muscle mass (ASM) [14,16,17,18] and it is an important clinical parameter for the diagnosis of sarcopenia [19,20].

Another parameter that can be measured with BIA is the phase angle (PhA), which is the ratio between electrical resistance and reactance. PhA has received increasing attention because it is believed to be a proxy for water distribution (ratio of extracellular water (ECW) to intracellular water (ICW)) and body cell mass (BCM) [21]. Therefore, a high PhA suggests increased cellularity (e.g., more BCM than FFM), cellular integrity, and cellular functions [21]. PhA decreases with inflammation, malnutrition, and prolonged physical inactivity [22] and is generally associated with worse quality of life [23] and prognoses in various chronic diseases [24,25,26]. In older adults it is also an independent predictor of adverse clinical outcomes such as frailty, falls, and mortality. The 2019 EWGSOP consensus suggested evaluating the introduction of PhA as an index of muscle quality [1].

The aim of this study was to test the combined role of muscle ultrasound and bioimpedentiometry for the diagnosis of sarcopenia in a population of hospitalized older males and females.

## 2. Materials and Methods

### 2.1. Study Population

For this study a sample of 70 subjects (50% females) was recruited from May 2022 to September 2022. The sample included 10 young adults with normal weight and 60 hospitalized older (50% females, mean age 83.6 ± 5.7 years) enrolled in the Geriatric Department of the University Hospital of Verona with good level of autonomy and collaboration.

The criteria for exclusion from the study were subjects who were bedridden or had serious clinical conditions such as fractures in the lower limbs and/or severe peripheral edema, which made it impossible to carry out the measurements or invalidated the results of ultrasound or impedentiometry.

All subjects underwent clinical evaluation and subsequent parameter measurements for the study. In hospitalized patients, height, weight, and waist circumference were recorded during the first 3 days of hospitalization. The Barthel index score was collected at admission and body mass index (BMI) was calculated as the ratio between weight and height squared (kg/m^2^), as reported elsewhere [6]. The study was approved by the Ethics Committee of Verona University (progressive number 1829CESC, approved 28 June 2018, protocol number 46319).

### 2.2. Handgrip Test

Strength assessment of the dominant hand flexor muscles was performed using a Spark 160 portable dynamometer (Spark, Iowa City, IA, USA). The patient was positioned sitting with the arm stretched out along the side and the elbow flexed at 90°, after which the grip strength of the hand was calculated [27]. For each subject three measurements were performed and the best of the three was considered. The cut-off used to evaluate the decrease in strength was assigned based on previously proposed cut-offs of <27 kg for men and <16 kg for women [1].

### 2.3. Bioelectric Impedance Analysis

A portable multi-frequency impedentiometer, “Human Im Touch” (DS medica, Milan, Italy), was used. This instrument was used at a single frequency of 50 kHz to analyze several parameters, including fat mass (FM), fat mass percentage (FM%), lean mass (FFM), and PhA. The device was subsequently connected to a computer for the software processing of the above-mentioned parameters and included additional parameters such as skeletal muscle (SM) and skeletal muscle index (SMI).

The data were then analyzed using the software and from the achieved results it was possible to obtain further information using the Sergi equation [16], including skeletal appendicular muscles (ASM) and indexed skeletal appendicular muscles (ASMI). The ASMI cut-offs used for a diagnosis of sarcopenia were those reported in various studies, including that of the EWGSOP2 in 2019 (1), i.e. <7 kg/m^2^ for men and <5.5 kg/m^2^ for women.

Sarcopenia was defined as the presence of low muscle strength combined with low muscle mass according to the EWGSOP2 criteria [1].

### 2.4. Ultrasound Study

An Esaote My Lab X6 ultrasound scanner (Esaote SpA, Genoa, Italy), equipped with a broadband linear transducer probe L 3–11 and an axial resolution of approximately 0.2 mm, was used. A preset for system settings was established in order to improve reliability of image acquisition in B mode: a stable gain was used for all patients with the focus fixed in the middle of the rectus femoris. Two operators (AZ, GV) with previous experience in muscle ultrasound acquired the scanned images twice after placing patients in a supine position and calculated the average value of each variable considered, as reported with more details elsewhere [6].

The acquisition of the images was followed by their analysis using the designated ImageJ program and through a blind procedure. The rectus femoris cross-sectional area was scanned at half the distance between the antero-superior iliac spine and the upper edge of the patella in the basal condition, and afterwards maximal compression was obtained through the probe. The stiffness index was therefore expressed as the difference between the two distances divided by the base (thickness of the rectus femoris) and then multiplied by one hundred. The higher the reduction percentage, the greater the compressibility of the analyzed muscle was considered to be.

The basal acquisition of the rectus femoris allowed for the post-processing calculation of the area (CSA, cm^2^) and for the average echogenicity of the pixels (0–255) using a dedicated program, ImageJ (version 1.53; National Institute of Health, Bethesda, MD, USA). Finally, the thickness of the subcutaneous fat was calculated as the distance between the dermis and the superficial aponeurosis of the rectus femoris. The muscle quality index (MQI) was calculated as the product of rectus femoris CSA measured by ultrasound and PhA measured by BIA.

### 2.5. Evaluation of Nutritional Status

At the initial visit, nutritional status was assessed using the Mini Nutritional Assessment (MNA) as reported elsewhere [28].

The original version of the tool which has been used in this study contained 18 weighted questions, divided into four nutritional areas including anthropometric measurements, a global assessment, a dietary assessment, one question on self-perception of whether food intake is sufficient, and one on self-experienced health status. The responses can give a maximum of 30 points.

### 2.6. Statistical Analysis

The study population was divided by sex and age. A *t*-test for unpaired data was used to compare demographic, anthropometric, bioimpedance measurement, and ultrasound characteristics in younger and older subjects by sex and in sarcopenic and non-sarcopenic older adults.

A linear regression was performed that considered physical strength measured by handgrip and ASMI as dependent variables and CSA, Barthel index, sex, BMI, and PhA as independent variables.

Finally, receiver operating characteristic (ROC) curves were made. This measure was obtained by relating the estimates of true positives (sensitivity) and the percentage of false positives (1-specificity) for each possible score that was obtained with ultrasound measurements for CSA, as well as with impedance measurements for PhA to obtain optimal cut-offs for the diagnosis of sarcopenia.

The area under the curve (AUC) was calculated as a measure of the diagnostic accuracy associated with different measurement values. Confidence intervals for AUC and comparison of ROC curves were calculated according to Delong et al. [29].

The cut-off values that maximized the relationship between sensitivity and specificity were obtained by calculating the Youden index. This index was calculated for all the points of the ROC curves: the threshold values for both sexes of CSA, of PhA, and of MQI were identified.

## 3. Results

### 3.1. Main Results

#### 3.1.1. Study Characteristics

The study population included 10 young adults with mean age 30 ± 1.6 years old (50% females) and 60 hospitalized older subjects with mean age 83.6 ± 5.7 years (50% females). Table 1 shows study characteristics of men and women according to age groups.

Young adults of both sexes showed higher handgrip strength, skeletal muscle mass, and PhA and lower percentage of fat mass as compared with hospitalized older subjects. In addition, in young adult males, lean mass was higher compared with the older counterpart.

Moreover, in both sexes, young adults, as compared with the hospitalized older group, showed higher CSA values, muscle thickness, and MQI and lower echogenicity. Young males also showed significantly lower compressibility than older adults. Hospitalized older participants (*n* = 60) were further divided into sarcopenic and non-sarcopenic based on ASMI cut-offs (ASM/h2) according to EWGSOP2 criteria [1,16], as shown in Table 2. Sarcopenia prevalence in the older study population was 43.3%. Lower MNA, PhA, FM, and FM% were observed in sarcopenic patients as compared with non-sarcopenic patients. In addition, lower SM, SMI, CSA, muscle thickness, and MQI and higher compressibility were observed in sarcopenic as compared with non-sarcopenic subjects.

#### 3.1.2. Multiple Regression Analysis

From multiple regression analyses that evaluated handgrip strength as a dependent variable, a direct association was observed with CSA, explaining 42.6% of the variance (β st = 0.456 *p* < 0.001), with the Barthel index (β st = 0.313 *p* = 0.001) and sex (β st = 0.278 *p* = 0.008), which explained approximately an additional 8% and 7% of the variance, respectively.

Considering ASMI as a dependent variable and CSA, BMI, PhA, and sex as independent variables, CSA was found to be a predictor (β st = 0.194 *p* < 0.001), explaining 56% of the variance, as were BMI (β st = 0.485 *p* < 0.001) and sex (β st = 0.396 *p* < 0.001), which explained an additional 15% and 12% of the variance, respectively. PhA (β st = 0.177 *p* < 0.001) measured by BIA explained an additional 2% of the variance.

#### 3.1.3. ROC Curves and Optimal Cut-Off Calculation

The area under the curve (AUC) relative to the diagnosis of sarcopenia for CSA, PhA, and MQI was calculated. The CSA values showed an AUC of 0.852 and 0.867 for females and for males, respectively. Calculation of the ROC curves of PhA showed an AUC of 0.792 in women and 0.898 in men. In order to find a parameter that had greater predictive power, ROC curves of MQI (the product of CSA and PhA) were analyzed. The obtained AUC was 0.900 for women and 0.969 for men (Figure 1).

In order to identify the optimal cut-off values of the parameters of CSA, PhA, and MQI for a diagnosis of sarcopenia, the Youden index was calculated (Table 3). The CSA values that obtained the best sensitivity and specificity values in our population of hospitalized subjects was 4.248 cm^2^ and 5.637 cm^2^, for women and for men, respectively. For PhA the values with the highest sensitivity and specificity were 3.85° and 4.25°, for women and for men, respectively. Finally, the MQI value with the best sensitivity and specificity was 15.63 cm^2^° for women and 23.45 cm^2^° for men (Table 3).

## 4. Discussion

The results of this study showed that hospitalized older men and women had a statistically significant decrease in CSA and muscle thickness as compared with young adults. A previous study by Strasser et al. [30] was consistent with our findings and showed an age-related decrease in muscle thicknesses of all muscles of the quadriceps femoris, especially in the rectus femoris muscle with a reduction in thickness of 25.4% [30]. Similarly, Narici et al., using both CT and ultrasonography, evaluated the effect of aging on the architecture of the gastrocnemius muscle between young and old subjects and concluded that CSA, muscle volume, fascicle length, and pennation angle were significantly lower in older subjects [31].

### 4.1. Body Composition Findings

We observed that older adults showed higher waist circumference, BMI, and fat mass percentage in comparison with young counterparts. Waist circumference is a surrogate of visceral adipose tissue that is known to increase with age, and it is widely reported that this depot is strictly related to intermuscular adipose tissue, which is the fat interspersed between muscle bundles [32]. This fat depot promotes lipotoxicity, which induces and aggravates insulin resistance, inflammation, mitochondrial dysfunction, and oxidative stress, all of which explain the link between central fat distribution and sarcopenia [33].

In older adults, malnutrition, characterized by low calorie intake, weight loss, and low BMI, often leads to sarcopenia onset [34]. In fact, in our study the older adult population with sarcopenia showed lower MNA scores as compared with non-sarcopenic subjects and also lower BMI, fat mass, fat mass percentage, and waist circumference. This is in line with a recent study from Curtis et al. which showed that underweight BMI, but not overweight or obesity, is significantly associated with increased odds of probable sarcopenia [35]. On the contrary, they showed that waist circumference was not significantly associated with probable sarcopenia [35].

### 4.2. Ultrasound Findings 

Comparison between sarcopenic and non-sarcopenic hospitalized older subjects showed that sarcopenic patients had significantly lower CSA and muscle thickness. Similarly, Minetto et al. in 2015 [36] observed a reduction in rectus femoris muscle thickness in sarcopenic patients as compared with healthy older patients and proposed a cut-off of 20 mm in males and 16 mm in females as threshold values for the diagnosis of sarcopenia.

In our study, older hospitalized subjects of both sexes showed significantly increased echogenicity values compared with younger subjects. This is in line with previous studies by Fukumoto et al. [37] which showed in a population of 92 women that echogenicity is directly related to age and inversely related to leg extensor strength.

Furthermore, in our study, there was a statistically significant difference in compressibility when comparing older hospitalized subjects with younger counterparts in both men and women. With aging, the increase in type I muscle fibers with higher lipid content [38] and the reduction in type II fibers [39,40] may partly explain the decrease in muscle density and stiffness. Moreover, as previously reported, adipose infiltration and fibrosis, which both increase with age, could influence muscle compressibility and echogenicity [6,41].

In the present study, comparing sarcopenic and non-sarcopenic hospitalized older subjects, no significant difference in echogenicity was found. This result is apparently in contrast to that obtained by Ismail et al. [42] wherein they found significantly lower muscle echogenicity values in 10 women with lower lean mass as compared with 10 with lean mass in the normal range. In the same study, echogenicity was inversely related to peak force. However, it should be emphasized that our patients had a narrow age range and had lean mass values at the lower limits of normality as compared to the Ismail et al. study population. Moreover, the role of echogenicity in defining muscle quality is still under debate.

Several studies have evaluated muscle quality using computerized grayscale analysis and explained that echogenicity correlates more with fibrous tissue than adipose tissue and that this relationship is to a lesser extent influenced by muscle quantity [15,43,44].

### 4.3. Bioimpedentiometry Findings

The results of our study showed that older female inpatients show a statistically significant decrease in PhA and skeletal muscle mass as compared with younger women. This trend was also seen in the male population. These results are consistent with those of Kolodziej et al. in 2019 [45], in which they demonstrated a significant difference in reactance and PhA measures between people older than 65 years and subjects at least 10 years younger. When we compared sarcopenic subjects with non-sarcopenic subjects, the latter were found to have on average a greater PhA and skeletal muscle mass than the former.

In a previous study by Marini et al. [46] with community-dwelling older people, it was found that PhA was markedly lower in sarcopenic subjects than in non-sarcopenic subjects (18.0% and 14.8% lower in men and women, respectively). Similarly, Espirito Santo Silva et al. in 2019 showed that sarcopenic subjects (12.6% of the total sample) had 22.4% lower PhA values than non-sarcopenic subjects [47].

Lastly, the performed multiple regression analysis allowed us to evaluate how CSA, BMI, Barthel scale, and PhA influence muscle strength and ASMI. We observed that CSA is associated with handgrip strength and ASMI. This confirms the results from Deniz et al. [48] that showed a correlation between CSA and grip strength in a population of patients with sarcopenic obesity. Similarly, Strasser [30] also found a correlation between rectus femoris muscle thickness and grip strength.

### 4.4. Combined Index MQI

In our study, through the use of ROC analysis, we calculated the cut-offs to identify values of CSA, PhA, and MQI that could be potentially used to diagnose the presence of sarcopenia. Our results are similar to those from Ozturk et al. [49] who measured CSA in two samples of geriatric patients (sarcopenic and non-sarcopenic subjects according to BIA analysis) and identified a cut-off of 5.2 cm^2^ for men and 4.3 cm^2^ for women.

In the present study, the PhA cut-off for the diagnosis of sarcopenia was found to be 3.850° (sensitivity 57.9%; specificity 100%) in women and 4.250° (sensitivity 73.3%; specificity 67.7%) in men. Our results are comparable to those of Yamada et al. [50] which, in a population of 1009 community-dwelling individuals, reported that the best cut-off values were 4.05° in men and 3.55° in women. Similarly, in a more recent study by Kosoku et al. [51], the authors found that the optimal PhA cut-off value for sarcopenia in kidney transplant recipients of both sexes is 4.46°, with a sensitivity of 74% and specificity of 70%. The low PhA values observed in our population may be explained by the fact that the present study involved hospitalized older adults characterized by moderate to low autonomy level, as evidenced by the Barthel index values [22], and by high age which is associated presumably with inflammation, malnutrition, and comorbidities [24,25,26].

Subsequently, we tested the combined predictive ability of ultrasound and bioimpedentiometry, calculating a new index, the MQI, the product of CSA and PhA; finally, ROC analysis was performed.

The threshold values of MQI were 15.63 (sensitivity 78.9%; specificity 100%) and 23.45 (sensitivity 100% and specificity 86.7%) in women and in men, respectively. AUC values were found to be 0.900 and 0.969 in women and men, respectively. Thus, in both men and women, the combined index appears to have a high predictive power for sarcopenia diagnosis as compared with the use of CSA or PhA alone.

### 4.5. Study Limitations

Some limitations warrant a mention. Firstly, for the diagnosis of sarcopenia, bioimpedance was used instead of more sophisticated techniques, such as computed tomography, magnetic resonance, or dual X-ray energy absorptiometry, considered the gold standard for sarcopenia diagnosis. However, BIA has now been included as a recognized method for muscle mass quantification [2]. Secondly, the number of healthy young controls was limited. Thirdly, this is a pilot study involving a small sample size of older adults and therefore our results must be considered with caution. In this regard, it will be necessary to replicate the study in larger populations to be able to confirm whether CSA and PhA can be used in combination as additional parameters to detect low muscle quality and to identify sarcopenia in hospitalized patients.

## 5. Conclusions

In conclusion, our study shows that ultrasound and bioimpedance measurements are reliable methods for studying muscle mass and quality, which is in line with previous studies and that the MQI, derived from a combination of echographic and impedentiometric characteristics, is a promising index for sarcopenia diagnoses in both male and female subjects.

## Figures and Tables

**Figure 1 nutrients-16-02429-f001:**
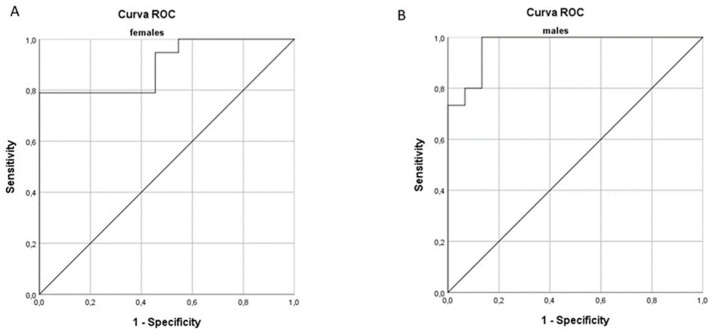
Receiver operating characteristic curve (ROC) for prediction of sarcopenia based on the muscle quality index (MQI) in females (**A**) and males (**B**).

**Table 1 nutrients-16-02429-t001:** Comparisons between young adults and the hospitalized older subjects divided by sex.

	Females (*n* = 35; 50%)				Males (*n* = 35; 50%)			
	Total Sample (*n* = 35)	Hospitalized Elderly Women (*n* = 30)	Young Females (*n* = 5)	*p* Value	Total Sample (*n* = 35)	Hospitalized Elderly Men (*n* = 30)	Young Males (*n* = 5)	*p* Value
Age (Years)	77.69 ± 20.48	85.67 ± 5.49	29.80 ± 0.66	<0.001	74.26 ± 18.87	81.6 ± 5.17	30.20 ± 1.92	<0.001
Height (cm)	159.11 ± 6.96	158.07 ± 5.99	165.40 ±9.66	0.027	171.97 ± 8.16	170.13 ± 7.2	183 ± 3.67	<0.001
Weight (kg)	57.45 ± 9.34	58.06 ± 9.66	53.8 ± 6.79	0.352	73.96 ± 16.03	73.65 ± 17.11	75.8 ± 7.6	0.786
Body Mass Index (kg/m^2^)	22.79 ± 3.50	23.31 ± 3.47	19.64 ± 1.61	0.028	24.52 ± 4.36	24.79 ± 4.55	22.92 ± 2.8	0.384
Waist Circumference (cm)	85.86 ± 14.03	90.37 ± 9.02	58.8 ± 3.96	<0.001	92.86 ± 14.39	96.23 ± 12.87	72.60 ± 8.23	<0.001
Grip Strength (kg)	15.89 ± 6.25	14.33 ± 5.17	25.22 ± 3.38	<0.001	26.16 ± 12.98	21.68 ± 7.11	53.06 ± 3.27	<0.001
Phase Angle (°)	3.87 ± 1.05	3.59 ± 0.81	5.58 ± 0.57	<0.001	4.33 ± 1.4	4.06 ± 0.85	6.96 ± 0.43	<0.001
Fat-Free Mass (kg)	36.83 ± 5.09	36.44 ± 5.15	39.18 ± 4.47	0.271	52.19 ± 10.27	50.26 ± 9.8	63.76 ± 2.35	0.005
Fat Mass (kg)	21.05 ± 8.17	22.12 ± 8.19	14.62 ± 4.58	0.056	21.91 ± 11.82	23.42 ± 11.81	12.84 ± 7.54	0.063
Fat Mass %	35.73 ± 0.88	37.21 ± 8.4	26.82 ± 6.17	0.013	28.49 ± 11.4	0.31 ± 0.11	0.16 ± 0.08	0.007
Skeletal Muscle (kg)	16.2 ± 3.98	15.64 ± 3.9	19.53 ± 2.64	0.041	27.54 ± 5.94	26.48 ± 5.73	33.89 ± 1.6	0.008
Skeletal Muscle Index (kg/m^2^)	6.36 ± 1.32	6.23 ± 3.98	7.14 ± 0.86	0.154	9.27 ± 1.8	9.131 ± 1.9	10.13 ± 0.59	0.257
Basal Thickness (cm)	1.25 ± 0.32	1.18 ± 0.25	1.65 ± 0.39	0.001	1.57 ± 0.46	1.44 ± 0.34	2.34 ± 0.28	<0.001
Post Compression Thickness (cm)	0.70 ± 0.24	0.65 ± 0.2	1.0 ± 0.26	0.001	0.94 ± 0.37	0.83 ± 0.23	1.61 ± 0.35	<0.001
Compressibility (%)	0.44 ± 0.11	0.45 ± 0.11	0.39 ± 0.11	0.251	0.41 ± 0.1	0.42 ± 0.08	0.31 ± 0.12	0.021
Stiffness (0–100)	55.88 ± 11.4	54.99 ± 11.47	61.22 ± 10.53	0.264	59.69 ± 9.67	58.18 ± 8.44	68.79 ± 12.5	0.021
Subcutaneus Fat (mm)	1.08 ± 0.34	1.11 ± 0.35	0.96 ± 0.22	0.358	0.72 ± 0.27	0.74 ± 0.28	0.61 ± 0.23	0.319
Rectus Femoris Cross-sectional Area (cm^2^)	5.01 ± 1.86	4.49 ± 1.25	7.16 ± 1.11	<0.001	7.22 ± 3.15	6.19 ± 1.80	12.65 ± 2.57	<0.001
Echogenicity (0–250)	55.65 ± 18.40	60.63 ± 14.49	25.81 ± 7.40	<0.001	46.63 ± 11.81	46.09 ± 9.78	28.84 ± 13.18	0.001
Muscle Quality Index (cm)	20.05 ± 11.12	16.73 ± 7.59	39.93 ± 7.54	<0.001	34.69 ± 24.84	25.81 ± 10.59	87.88 ± 17.46	<0.001

**Table 2 nutrients-16-02429-t002:** Comparisons between sarcopenic and non-sarcopenic hospitalized old subjects (*n* = 60).

	Sarcopenic Hospitalized Elderly (*n* = 26)	Non-Sarcopenic Hospitalized Elderly (*n* = 34)	*p*-Value
Age (Years)	83.81 ± 5.89	83.50 ± 5.58	0.837
Sex (F%)	11 (42.3)	19 (55.8)	0.435
Barthel Index	54.0 (27.9)	68.5 (31.2)	0.067
Height (cm)	164.10 ± 8.95	163.27 ± 10.18	0.534
Weight (kg)	59.02 ± 13.21	71.09 ± 15.89	0.003
Body Mass Index (kg/m^2^)	21.98 ± 3.18	25.63 ± 4.02	<0.001
Waist Circumference (cm)	87.50 ± 7.48	97.74 ± 11.43	<0.001
Grip Strength (kg)	14.34 ± 5.62	20.81 ± 7.06	<0.001
Mini Nutritional Assessment	21.42 ± 7.33	25.56 ± 7.61	0.037
Phase Angle (°)	3.33 ± 0.61	4.20 ± 0.83	<0.001
Fat-Free Mass (kg)	41.07 ± 9.84	45.09 ± 10.67	0.14
Fat Mass (kg)	18.12 ± 7.22	26.33 ± 10.62	0.001
Fat Mass %	0.31 ± 0.08	0.36 ± 0.11	0.023
Skeletal Muscle (kg)	18.71 ± 6.43	22.86 ± 7.53	0.028
Skeletal Muscle Index (kg/m^2^)	6.90 ± 1.95	8.28 ± 2.20	0.014
Basal Thickness (cm)	1.22 ± 0.36	1.40 ± 0.29	0.036
Post Compression Thickness (cm)	0.62 ± 0.21	0.83 ± 0.20	<0.001
Compressibility (%)	0.49 ± 0.09	0.39 ± 0.09	<0.001
Stiffness (0–100)	51.27 ± 9.34	60.65 ± 8.80	<0.001
Subcutaneus Fat (mm)	0.89 ± 0.40	0.95 ± 0.33	0.589
Rectus Femoris Cross-sectional Area (cm^2^)	4.42 ± 1.55	6.04 ± 1.60	<0.001
Echogenicity (0–250)	51.12 ± 12.71	55.08 ± 15.34	0.291
Muscle Quality Index (cm)	14.92 ± 6.28	26.13 ± 10.04	<0.001

**Table 3 nutrients-16-02429-t003:** Cross-sectional area, phase angle, and muscle quality index cut-off values obtained by calculating the Youden index.

Females							Males					
**Cross-sectional Area (cm^2^)**	<3.49	<3.61	<3.71	<3.88	<4.08	<4.24 ^a^	<2.69	<3.59	<4.28	<4.88	<5.49	<5.63 ^a^
**Sensitivity (%)**	94.7	89.5	84.2	78.9	78.9	78.9	100	100	100	100	100	100
**Specificity (%)**	45.5	54.5	54.5	63.6	81.8	100	6.7	20	33.3	46.7	60	80
**Phase Angle (°)**	<2.25	<2.55	<2.85	<3.35	<3.6	<3.85 ^a^	<2.6	<3.05	<3.35	<3.7	<3.95	<4.25 ^a^
**Sensitivity (%)**	94.7	89.5	84.2	78.9	68.4	57.9	100	100	100	93.3	73.3	73.3
**Specificity (%)**	9.1	18.2	27.3	63.6	72.7	100	6.7	20	46.7	66.7	73.3	93.3
**Muscle Quality Index**	<6.78	<8.50	<10.24	<11.39	<13.23	<15.63 ^a^	<8.35	<11.29	<13.38	<17.37	<22.20	<23.45 ^a^
**Sensitivity (%)**	100	94.7	84.2	78.9	78.9	78.9	100	100	100	100	100	100
**Specificity (%)**	9.1	27.3	36.4	54.3	72.7	100	6.7	20	33.3	46.7	66.7	86.7

^a^ Optimal values based on ROC analysis.

## Data Availability

The data underlying this article cannot be shared publicly as this option was not included in the informed consent form signed by participants. The data will be shared on reasonable request to the corresponding author.

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
