# Peer review of "The Role of Combined Muscle Ultrasound and Bioimpedentiometry Parameters for Sarcopenia Diagnosis in a Population of Hospitalized Older Adults"

_nutrients, 2024, doi:10.3390/nu16152429_

Round 1

Reviewer 1 Report

Comments and Suggestions for Authors

In the current study, the authors evaluated the efficacy of the combined muscle ultrasound and bioimpedance techniques for diagnosing sarcopenia in hospitalized older males and females. This is a well-done and well-written study. The authors described well the limitations of the study, including the small sample size.

One suggestion that I have is to discuss the findings of the significant difference in waist circumference between old and young subjects and how it could affect sarcopenia.  It was interesting that the non-sarcopenic subjects had larger waist circumference than the sarcopenic. It would be interesting to know the author's opinions on these differences.

Author Response

Response to Reviewer 1 Comments

1. Summary

We would like to thank the Reviewer for taking the time to review this manuscript. Please find the detailed responses below and the corresponding revisions highlighted changes in the re-submitted files.

2. Questions for General Evaluation

Reviewer’s Evaluation

Response and Revisions

Does the introduction provide sufficient background and include all relevant references?

Yes/Can be improved/Must be improved/Not applicable

Are all the cited references relevant to the research?

Yes/Can be improved/Must be improved/Not applicable

Is the research design appropriate?

Yes/Can be improved/Must be improved/Not applicable

Are the methods adequately described?

Yes/Can be improved/Must be improved/Not applicable

Are the results clearly presented?

Yes/Can be improved/Must be improved/Not applicable

Are the conclusions supported by the results?

Yes/Can be improved/Must be improved/Not applicable

3. Point-by-point response to Comments and Suggestions for Authors

Comments 1: In the current study, the authors evaluated the efficacy of the combined muscle ultrasound and bioimpedance techniques for diagnosing sarcopenia in hospitalized older males and females. This is a well-done and well-written study. The authors described well the limitations of the study, including the small sample size.

Response 1: Response 1: we thank the Reviewer for his/her comments and for the concerns raised, which helped us to improve the quality of the manuscript.

Comments 2: One suggestion that I have is to discuss the findings of the significant difference in waist circumference between old and young subjects and how it could affect sarcopenia.

Response 2: We agree with the Reviewer’s comment and we have, accordingly, modified the discussion to emphasize this point.

Page 10 Line 282

We observed that older adults showed higher waist circumference, BMI and fat mass percentage in comparison with the young counterparts. Waist circumference is a surrogate of visceral adipose tissue that is known to increase with age, and it is widely reported that this depot is strictly related to intermuscular adipose tissue, which is the fat interspersed between muscle bundles [33]. This fat depot promotes lipotoxicity, which induces and aggravates insulin resistance, inflammation, mitochondrial dysfunction and oxidative stress, all of which explain the link between central fat distribution and sarcopenia [34].

33.       Gallagher D, Kuznia P, Heshka S, Albu J, Heymsfield SB, Goodpaster B, Visser M, Harris TB. Adipose tissue in muscle: a novel depot similar in size to visceral adipose tissue. Am J Clin Nutr. 2005; 81(4): 903-10.

34.       D’Souza K, Mercer A, Mawhinney H, Pulinilkunnil T, Udenigwe CC, Kienesberger PC. Whey peptides stimulate differentiation and lipid metabolism in adipocytes and ameliorate lipotoxicity-induced insulin resistance in muscle cells. Nutrients. 2020; 12:425.

Comments 3: It was interesting that the non-sarcopenic subjects had larger waist circumference than the sarcopenic. It would be interesting to know the author's opinions on these differences.

Response 3: We agree with the Reviewer comment and we have, accordingly, modified the discussion to emphasize this point.

Page 10 Line 291

In older adults, malnutrition, characterized by low calorie intake, weight loss and low BMI, often lead to sarcopenia onset [35]. In fact, in our study the older adult population with sarcopenia showed lower MNA score as compared with non-sarcopenic subjects and also lower BMI, fat mass, fat mass percentage and waist circumference. This is in line with a recent study from Curtis which showed that underweight BMI, but not overweight or obesity, is significantly associated with increased odds of probable sarcopenia [36]. On the contrary, they showed that waist circumference was not significantly associated with probable sarcopenia [36].

  1. Nishioka, S. Current Understanding of Sarcopenia and Malnutrition in Geriatric Rehabilitation. Nutrients 2023, 15, 1426.
  2. Curtis M, Swan L, Fox, R, Warters A, O’Sullivan M. Associations between Body Mass Index and Probable Sarcopenia in Community-Dwelling Older Adults. Nutrients 2023, 15, 1505.

Reviewer 2 Report

Comments and Suggestions for Authors

Thank you for the opportunity to review your manuscript, “The role of combined ultrasound muscle and bioimpedentiometry parameters for sarcopenia diagnosis in a population of hospitalized older adults”

The aim of the study was to test the combined role of muscle echography and bioimpedentiometry for the diagnosis of sarcopenia in a population of hospitalized older males and females.

The manuscript is clear and well laid out, with easy-to-follow writing.

Some minor aspects that may help improve the manuscript:

In the introduction, transitions between sentences and paragraphs should be more fluid. For example, after discussing muscle quality, the methods (BIA and ultrasound) are smoothly transitioned to methods (BIA and ultrasound), highlighting their relevance for assessing muscle parameters.

At the end of the introduction, ensure the language used is precise and scientifically rigorous. For example, instead of "muscle echography," consider using "muscle ultrasound" or "echogenicity measurements."

Specify the criteria for selecting the 70 subjects in the study more explicitly, mainly how the sample size was determined. It's essential to justify why this sample size was sufficient for the study's objectives and statistical analyses.

Expand slightly on how the Mini-Nutritional Assessment (MNA) was integrated into the study, especially regarding its role in assessing potential confounders or moderators of sarcopenia and if it is validated in the older adult population.

Consider structuring the discussion into subsections corresponding to the study's main findings (e.g., ultrasound findings, bioimpedentiometry findings, combined index MQI, and comparison with previous studies). This will improve clarity and help readers navigate through the different aspects of the discussion more easily.

Author Response

Response to Reviewer 2 Comments

1. Summary

We thank the Reviewer for taking the time to review this manuscript. Please find the detailed responses below and the corresponding revisions highlighted changes in the re-submitted files.

2. Questions for General Evaluation

Reviewer’s Evaluation

Response and Revisions

Does the introduction provide sufficient background and include all relevant references?

Yes/Can be improved/Must be improved/Not applicable

Are all the cited references relevant to the research?

Yes/Can be improved/Must be improved/Not applicable

Is the research design appropriate?

Yes/Can be improved/Must be improved/Not applicable

Are the methods adequately described?

Yes/Can be improved/Must be improved/Not applicable

Are the results clearly presented?

Yes/Can be improved/Must be improved/Not applicable

Are the conclusions supported by the results?

Yes/Can be improved/Must be improved/Not applicable

3. Point-by-point response to Comments and Suggestions for Authors

Comments 1: Thank you for the opportunity to review your manuscript, “The role of combined ultrasound muscle and bioimpedentiometry parameters for sarcopenia diagnosis in a population of hospitalized older adults”

The aim of the study was to test the combined role of muscle echography and bioimpedentiometry for the diagnosis of sarcopenia in a population of hospitalized older males and females.

The manuscript is clear and well laid out, with easy-to-follow writing.

Response 1: we thank the Reviewer for its comment and for the concerns raised which helped us to improve the quality of the manuscript.

Comments 2: Some minor aspects that may help improve the manuscript:

In the introduction, transitions between sentences and paragraphs should be more fluid. For example, after discussing muscle quality, the methods (BIA and ultrasound) are smoothly transitioned to methods (BIA and ultrasound), highlighting their relevance for assessing muscle parameters.

Response 2: the introduction has been revised in accordance with reviewer comment.

Comments 3: At the end of the introduction, ensure the language used is precise and scientifically rigorous. For example, instead of "muscle echography," consider using "muscle ultrasound" or "echogenicity measurements."

Response 3: We agree with the Reviewer comment and we have modified the introduction accordingly.

•Comments 4: Specify the criteria for selecting the 70 subjects in the study more explicitly, mainly how the sample size was determined. It's essential to justify why this sample size was sufficient for the study's objectives and statistical analyses.

Response 4: We agree with the Reviewer’s comment, but this is a pilot study and the number of subjects is based on another study from Puthucheary using muscle ultrasound to evaluate skeletal muscle wasting in survivors of critical illness. This study was performed in 63 patients similar to our study. (Puthucheary ZA, Rawal J, McPhail M, et al. Acute Skeletal Muscle Wasting in Critical Illness. JAMA. 2013;310(15):1591–1600. doi:10.1001/jama.2013.278481)

We decided to modify the study limitations in order to highlight this point better.

Page 12 line 382

Secondly, the number of healthy young controls was limited. Thirdly, this is a pilot study involving a small sample size of old adults and therefore our results must be considered with caution.

Comments 5: Expand slightly on how the Mini-Nutritional Assessment (MNA) was integrated into the study, especially regarding its role in assessing potential confounders or moderators of sarcopenia and if it is validated in the older adult population.

Response 5: We agree with the Reviewer comment and we have, accordingly, modified the discussion to emphasize this point.

Page 10 Line 291

In older adults, malnutrition, characterized by low calorie intake, weight loss and low BMI, often lead to sarcopenia onset [35]. In fact, in our study the older adult population with sarcopenia showed lower MNA score as compared with non-sarcopenic subjects and also lower BMI, fat mass, fat mass percentage and waist circumference.  

•Comments 6: Consider structuring the discussion into subsections corresponding to the study's main findings (e.g., ultrasound findings, bioimpedentiometry findings, combined index MQI, and comparison with previous studies). This will improve clarity and help readers navigate through the different aspects of the discussion more easily.

Response 6: We agree with the Reviewer comment and we have, accordingly, modified the discussion including subsections for  study’s main finding in order to improve readability.

Round 2

Reviewer 2 Report

Comments and Suggestions for Authors

I want to congratulate the authors for their work.

The authors have answered all my questions and made appropriate changes to the manuscript. 

Lines 316-321 are in italics. I understand that this is a typographical error.